# Reptile Host Associations of *Ixodes scapularis* in Florida and Implications for *Borrelia* spp. Ecology

**DOI:** 10.3390/pathogens10080999

**Published:** 2021-08-07

**Authors:** Carrie De Jesus, Chanakya Bhosale, Kristen Wilson, Zoe White, Samantha M. Wisely

**Affiliations:** Department of Wildlife Ecology and Conservation, University of Florida, Gainesville, FL 32603, USA; carriedejesus@ufl.edu (C.D.J.); cbhosale@ufl.edu (C.B.); knwilson@ufl.edu (K.W.); zseganish@ufl.edu (Z.W.)

**Keywords:** Lyme disease, *Borrelia burgdorferi sensu lato*, host association, lizard

## Abstract

Host associations of the tick vector for Lyme Borreliosis, *Ixodes scapularis*, differ across its geographic range. In Florida, the primary competent mammalian host of Lyme disease is not present but instead has other small mammals and herpetofauna that *I. scapularis* can utilize. We investigated host–tick association for lizards, the abundance of ticks on lizards and the prevalence of *Borrelia burgdorferi sensu lato* (*sl*). To determine which lizard species *I. scapularis* associates with, we examined 11 native lizard species from historical herpetological specimens. We found that (294/5828) of the specimens had attached ticks. The most infested species were *Plestiodon* skinks (241/1228) and *Ophisaurus* glass lizards (25/572). These species were then targeted at six field sites across Florida and sampled from June to September 2020, using drift fence arrays, cover boards and fishing. We captured 125 lizards and collected 233 immature *I. scapularis*. DNA was extracted from ticks and lizard tissue samples, followed by PCR testing for *Borrelia* spp. Of the captured lizards, 69/125 were infested with immature *I. scapularis*. We did not detect *Borrelia* spp. from tick or lizard tissue samples. Overall, we found that lizards are commonly infested with *I. scapularis*. However, we did not detect *Borrelia burgdorferi sl*. These findings add to a growing body of evidence that lizards are poor reservoir species.

## 1. Introduction

In the US Lyme Borreliosis (LB) is the most commonly reported vector-borne disease with approximately 476,000 cases per year [1]. The primary etiological agent of LB in the US is *Borrelia burgdorferi sensu stricto* (*ss*). *Borrelia burgdorferi ss* is part of a global species complex referred to as *Borrelia burgdorferi sensu lato* (*sl*) [2]. Species in the *B. burgdorferi sl* complex vary in their genetic code, pathogenicity, tick vectors, host species and distribution [3,4]. In the US, 10 species in this complex have been reported and are vectored by *Ixodes* ticks [4]. The distribution of these bacterial species has important implications on LB disease risk since they are not all pathogenic. In the US, two species in the complex are associated with LB: *B. burgdorferi ss* and *B. mayonii*. An additional species, *B. bissetiae*, is tentatively considered an agent of LB [2,3]. All three of the LB-associated genospecies are vectored by *Ixodes scapularis*. 

The distribution of *B. burgdorferi sl* species varies across North America. In California, *B. burgdorferis ss* was more frequently detected in the north and central coast, while *B. bissetiae* was more common in the southern part of the state [3]. In eastern North America, *B. burgdorferi sl* species prevalence is higher in the northeast and Midwest compared to the southeast [5,6,7]. Multiple *B. burgdorferi sl* species have been detected in the northeast and Midwest including *B. burgdorferi ss* and *B. mayonii* [2,6,7,8]. The distribution of *B. burgdorferi sl* species in the southeastern US, however, has not been well established since wildlife hosts and ticks have not been thoroughly assessed [9]. Understanding the tick and wildlife host associations can help provide insight into why there is a low prevalence of *B. burgdorferi sl* in the southeastern US. 

Multiple hypotheses have been proposed as to why the prevalence of *B. burgdorferi sl* species in hosts and ticks is concentrated in the Midwest and northeast but not in the southeast, even though *I. scapularis* is present in all regions. One hypothesis is that *I. scapularis* questing height can influence host associations. In the southeast, *I. scapularis* quest in the leaflitter, while in the northeast, they climb up vegetation, which could influence host associations [10]. An alternative hypothesis is that competent host availability may differ between regions. In the northeast, small mammals are the primary hosts of immature *I. scapularis* [11]. *Peromyscus leucopus* (white-footed mouse) is an extremely competent host of *I. scapularis* immatures and reservoir for *B. burgdorferi ss* [12]; however, *P. leucopus* is not present in the southern range of *I. scapularis’* distribution. Other rodent hosts are present in the southeastern US such as *P. gossypinus*, which can maintain long-term *B. burgdorferi ss* and *B. bissetiae* infections (>47 months) [13]; however, *P. gossypinus* are more commonly infested by *I. minor* and *I. affinis*, which rarely feed on humans compared to *I. scapularis* [13]. In the European *B. burgdorferi sl* complex system, different genotypes are associated with specific vertebrate taxa: mammals, birds and reptiles. However, the host association of different *B. burgdorferi sl* species in North America is not well established [4]. 

In the southeastern US, many alternative host species are available, on which *I. scapularis* can feed, including a diversity of lizard species. *Ixodes scapularis* immatures have been reported to feed on 14 reptile species in the US, and many of them reside in the southeast [14,15,16]. It has not been well studied which lizard species act as hosts for *I. scapularis*, since few studies examining wildlife for ticks incorporate lizards. In the few studies examining lizard hosts, the focus was placed on tick presence and infestation intensity; *Borrelia* infections have rarely been surveyed [9,16,17]. From these studies, the most commonly infested lizard species were *Plestiodon* skinks and *Ophisaurus* glass lizards [3,8,13,14]. These lizards are commonly found in the leaf litter, which is consistent with tick questing behavior in the southeast [6,16]. Even though multiple species of lizards can be infested with *I. scapularis*, they are the least studied host in *Borrelia* epidemiology and other tick-borne pathogens [9,11,18,19,20]. In the European *B. burgdorferi sl* complex, lizards are specifically the host of *B. lusitaniae*, a nonpathogenic species [3]. Whether or not lizards in the US are associated with specific *B. burgdorferi sl* species is unclear. 

Studies addressing *Borrelia* infections in lizards have had varying implications for the epidemiology of *Borrelia* transmission cycles. In California, the western fence lizard (*Sceloporus occidentalis*) plays a crucial role in the ecology of *Borrelia burgdorferi ss*. The blood of the western fence lizard contains a thermolabile, borreliacidal factor that kills *Borrelia* spirochetes [21]. Borreliacidals have not been identified in southeastern lizards, and there are reports of lizards infected with *Borrelia* [22,23]. Experimental infections using infected feeding ticks demonstrated that *Plestiodon inexpectatus* (southeastern five lined skink) and *Anolis carolinensis* (green anole) can acquire and maintain *Borrelia burgdorferi ss* infections [23]. Wild-caught lizards from South Carolina and north Florida have been found to be infected with species from the *B. burgdorferi sl* complex with a 50% prevalence from blood samples [22]. Another study of *Plestiodon* spp. collected across the southeast suggested that the skinks had zooprophylactic effect on *B. burgdorferi ss* infection [24]. In this study, *I. scapularis* nymphs infected with *B. burgdorferi ss* were allowed to feed on skinks. After feeding, the number of infected nymphs decreased from 71.4% to 7.4% [24]. The exact role of lizard hosts in the transmission of *Borrelia* is still ill-defined and likely complex. 

Florida is one of the most herpetologically diverse states [25], and *Ixodes scapularis* has been reported or has established populations in all 67 Florida counties. Cases of autochthonous LB have been reported in 45/67 Florida counties [26,27,28]. Studies that have examined reptile hosts of *I. scapularis* in Florida have only sampled within the northern part of the state [6,22,23,24], yet Florida constitutes the most southern range of the *I. scapularis* distribution in the US. How lizards influence the epidemiology of *Borrelia* in the most southern part of *I. scapularis* range has not been thoroughly investigated. Given Florida’s diverse lizard community, distribution of *I. scapularis* and LB cases, further investigations into the role of lizards in the epidemiology of *Borrelia* are warranted. 

To determine if lizards play a role in *Borrelia* epidemiology in Florida, we examined several different components of host competency. A competent reservoir host species must fulfill three criteria. First, a host and vector species must have interactions with one another. Are ticks feeding on lizards? Secondly, the host species must be able to maintain a pathogen infection. Are lizards infected with *Borrelia*? Lastly, the host must be able to infect a vector species. Can feeding ticks obtain the pathogen from feeding on lizards?

The objective of our investigation was to determine the role of native lizards in Florida for the first two criteria of host competency. We addressed the following questions. (1) Are *I. scapularis* ticks feeding on lizards in Florida? To address whether ticks and lizards interact, we examined museum specimens from historical and current collections for *I. scapularis* infestations. After determining tick–host associations of lizards we applied this knowledge to field sampling of lizards. We collected additional data on tick infestation from field-captured lizards. We then focused on the second aspect of host competency: (2) Are lizards infected with *Borrelia* or other tick-borne bacteria? Using field-collected ticks and tissue samples from captured lizards, we tested for the presence of *Borrelia* infections and other tick-borne bacterial pathogens.

## 2. Results

### 2.1. Museum Specimen Survey

We examined 11 species of native lizards, specimens of which were collected from all 67 counties. From the herpetological collection, we found that 5.0% (294/5828) of the specimens of all 11 species we examined were infested with ticks from 43 of the 67 Florida counties (Table 1). We found two tick species on the lizard specimens: *I. scapularis* and *Amblyomma americanum*.

*Sceloporus undulatus* was the only lizard to be infested with both tick species. Of the *S. undulatus* examined, 11 were infested with *I. scapularis* and 9 with *A. americanum*. None of the *S. undulatus* specimens were coinfested with both tick species. All other ticks identified on museum specimens were identified as immature *I. scapularis*. *Ixodes scapularis* were found on lizards which were collected as early as 1927 on *P. laticeps* specimens from Alachua County.

From the museum specimens, we found that 8/11 native lizard species were infested with ticks. *Plestiodon* was the most commonly infested lizard genus with ticks. *Plestiodon fasciatus, P. inexpectatus,* and *P. laticeps* had the greatest number of attached ticks per individual on average (Table 1). *Plestiodon egregius* was the only *Plestiodon* spp. to have only one specimen infested with ticks. *Ophisaurus* glass lizards also had tick infestations, although few individuals were infested compared to *Plestiodon* spp. 

### 2.2. Field Collected Lizards and Ticks

From our six field sites, we live captured a total of 125 lizards (Figure 1). We captured 123 individuals of *Plestiodon* spp. and two *O. ventralis* (Figure 1). The majority of captured lizards came from the two most northern sites, which mostly consisted of *P. fasciatus* and *P. laticeps*. 

*Plestiodon inexpectatus* was the most widely distributed lizard species; we collected them from five of six field sites. *Ophisaurus ventralis* were only captured from one central field site and not infested with ticks. We captured fewer lizards at the southern sites compared to the northern sites. 

Of the field collected lizards, 69/125 (55.2%) were infested with *I. scapularis* immatures (Table 2). In total we collected 233 (29 larvae, 204 nymphs) *I. scapularis* from the lizards. No other tick species were found on the field-collected lizards. We collected *I. scapularis* at five out of six field sites (Table 2). The majority of the ticks collected were from lizards in the northern field sites (Table 2). The central and southern field sites had lizards with fewer attached ticks. 

Three *Plestiodon* species were infested with ticks: *P. fasciatus*, *P. inexpectatus*, and *P. laticeps*, which was consistent with the museum specimen findings (Figure 1, Table 2). On average, *Plestiodon* spp. were infested with 3.4 ticks per individual (Table 3). Of the *Plestiodon* spp., we captured male, female, and juvenile individuals (Table 3). We found that *P. laticeps* had the greatest abundance of individuals infested with ticks compared to the other *Plestiodon* spp. (Table 2). All the *P. laticeps* individuals caught in our study were adults and all individuals were infested with ticks. *Plestiodon fasciatus* individuals were also commonly infested with ticks (66.7%). *Plestiodon inexpectatus* had the lowest average number of attached ticks and the lowest abundance of individuals infested with ticks (Table 3). 

We examined the abundance (no. of lizards infested with ticks/no. of lizards sampled) of tick-infested lizards across Florida from museum and field specimens (Figure 1). Multiple northern counties and parts of the panhandle had the highest abundance of lizards infested with ticks. The abundance of tick-infested lizards throughout central and southern Florida ranged from 0 to 0.20. Wakulla County in the Panhandle had the highest abundance of infested lizards at 0.4 ticks per individual. 

### 2.3. Pathogen Screening

We screened 103 tissue samples from the same 11 Florida native species from the Genetic Repository of the Florida Museum on Natural History (Table 1). No samples tested positive for any bacterial pathogens (*Ehrlichia* spp., *Anaplasma* spp., *Borrelia* spp., *Rickettsia* spp.).

We also screened field-collected tick and lizard tissue samples for tick-borne bacterial pathogens. We did not detect any *Borrelia* spp. in the tick or lizard tissue samples. Samples were also negative for the presence of *Anaplasma* spp. and *Ehrlichia* spp. Field-collected ticks tested positive for *Rickettsia* spp. for both the gltA and ompA gene. For the gltA gene primer set, we found that 14/69 tick pools (pooled by individual lizard) tested positive and matched closely with a *Rickettsia* spp. (GenBank Assession No. CP060138). With the ompA gene primers, we detected *Rickettsia* spp. in ticks pooled by individual (16/68). This sequence was similar to that of an *I. scapularis* rickettsial endosymbiont (GenBank Assession No. EF689735).

## 3. Discussion

The goal of our investigation was to address two aspects of host competency. (1) Do ticks and lizards interact? (2) Are lizards infected with *Borrelia burgdorferi sl*? 

To investigate if lizards and ticks interacted with each other in Florida, we surveyed historical herpetological specimens and field-captured lizards for tick infestations. We found that 5.0% of the herpetological specimens and 55.2% of field-collected lizards were infested with ticks. *Plestiodon* spp. and *Ophisaurus* spp. were the most common hosts of *I. scapularis* immatures throughout Florida. Our results are consistent with other host-association studies in the southeastern US for lizards [16,20,24]. One other study surveyed for ticks on museum specimens collected in North Carolina [19]. In that study, the authors found that 8.7% of the herpetological specimens were infested with *I. scapularis* immatures. While we found a similar abundance of infested ticks for *P. inexpectatus* as found in North Carolina (13.8%), *P. laticeps* (7.4%) and *P. fasciatus* (3.0%) in North Carolina had a lower abundance of infested ticks than in Florida specimens. 

*Anolis carolinensis* is a common lizard species throughout Florida and has been shown to be a competent reservoir host of *B. burgdorferi ss* in the laboratory [23]. Of the 1241 museum specimens collected from Florida, none were infested with ticks. In North Carolina, 3.2% (6/187) of the *A. carolinensis* examined were infested with larval ticks [23]. This apparent difference may have an ecological explanation driven by an invasive anole in Florida. *Anolis sagrei*, the brown anole, is a common invasive lizard species in Florida. In the presence of *A. sagrei*, *A. carolinensis* increases their arborealism [29]. These behavioral changes may reduce the likelihood of *A. carolinensis* interacting with *I. scapularis*, which quests in the leaf litter. Thus, it appears that *A. carolinensis* is not a likely competent host for *Borrelia* spp. given its lack of interaction with *I. scapularis*, in the presence of *A. sagrei*. 

It is important to note that tick–host association data from museum specimens must be interpreted with caution [19,30]. We found lower tick abundance in the museum specimens compared to the field specimens. Museum specimens are preserved after an animal has died. The time between death and preservation may allow ticks time to vacate their host and not be included with the preserved specimen, altering estimates of tick abundance and the intensity of individual infestation. Sampling bias, including the location, time, and sex of specimens, may also influence the occurrence of ticks on the collected specimen. In our museum specimens, Florida counties were not all equally represented in sampling efforts. We are likely missing data from certain populations throughout the state. Even with these biases, we were able to detect the occurrence of ticks infesting lizards in 43/67 of the Florida counties. 

Using the museum specimens, we were able to survey a wide diversity of lizards to determine which species were associated with Ixodid ticks. Museum specimens provide a convenient way to identify host species but are likely not as useful for comparing infestation rates or determining the absence of a tick species. Nonetheless, museum collections help clarify tick–host associations and can assist in validating those associations in the field. Studying wildlife infested with ticks requires various permitting processes and animal care requirements. By predetermining targeted host species with museum data, investigators can provide permitting agencies preliminary data on why a particular species needs to be surveyed. 

To complement our data from museum collections, we field-collected *Plestiodon* spp. to determine the magnitude of the interaction with *I. scapularis* in Florida. We found that 56.1% of the *Plestiodon* spp. we captured were infested with larval and nymphal *I. scapularis*. Most of our infested lizards came from two northern sites, where we collected mostly *P. fasciatus* and *P. laticeps*. The distribution of *P. fasciatus* and *P. laticeps* are both restricted to northern Florida and the Panhandle [23]. We caught *P. inexpectatus* across all our field sites as expected given its broad distribution across Florida. Although only a few individuals were captured at each site (n = 1–16), we found lizards with ticks at central and southern field sites. *Plestiodon inexpectatus* was found to be a competent host for *Borrelia* bacteria in a laboratory setting; yet in the field, they had the least number of ticks infesting them compared to the other *Plestiodon* spp. [23]. All three *Plestiodon* spp. are sympatric in forested habitats throughout the southeastern US [31]. Temperature influences how these three species partition themselves within their microhabitat [31]. *Plestiodon inexpectatus* can utilize warmer open habitats compared to the other *Plestiodon* spp. [31]. The use of warmer habitats may reduce possible tick interactions of *P. inexpectatus* since ticks are sensitive to desiccation [32].

Because lizards are common hosts of *I. scapularis* immature life stages in Florida, they may be important hosts for maintaining tick populations compared to northeastern or Midwest US [6]. *I. scapularis* in the southeastern US quest in leaf litter to avoid desiccation, as opposed to northern populations that climb up vegetation to quest [33,34]. *Plestiodon* spp. are common lizards that forage for arthropods in the leaf litter, which increases their likelihood of encountering *I. scapularis* [34,35]. Indeed, in California, lizard hosts are important for maintaining *I. pacificus* populations [36]. When lizards were removed from experimental plots, *I. pacificus* populations decreased [32]. However, *I. pacificus* feed preferentially on lizard hosts, which could have influenced the tight association of host and tick [36,37]. Previous studies of *I. scapularis* found that larvae preferred mice, but nymphal ticks did not display a preference [37,38]. Whether lizards are crucial for maintaining *I. scapularis* populations in the southeast has yet to be investigated. 

To address the second aspect of host competency, we screened lizard tissue samples for *Borrelia burgdorferi sl* to determine if *Borrelia* spp. were disseminated in host tissue. We did not detect *Borrelia burgdorferi sl* in any lizard tissue samples from our field or museum collection. Nor did we find *Borrelia* spp. in any of the ticks that we collected from lizards in the field. Although previous studies surveying ticks in the southeastern US found *Borrelia burgdorferi sl* at a low prevalence compared to the northeastern US [5,6,39], few studies have tested lizards and their attached ticks from field collected samples. While we can only speculate if our negative results were due to a lack of host competency, some lizard species have been previously hypothesized to have decreased competency due to a borreliacidal factor that decreases transmission of *Borrelia* spp. The borreliacidal factor has been shown to kill spirochetes in blood and sera samples of *Sceloporus occientalis* in western North America [21]. Xenodiagnositcs on *Plestiodon* spp. and *S. undulates* have been conducted and found that the number of infected ticks decreased after feeding on lizards [24,40,41]; however, the mechanism underlying decreased infection rates has not been identified. Xenodiagnostic studies apply infected immature ticks to lizards and then test the fed larvae for *Borrelia burgdorferi ss* infections. It is currently unknown if the *Plestiodon* species investigated in this study have the ability to curtail infections.

Tail snips are a commonly collected tissue type in herpetological studies. Tail snips are a simple tissue to sample since lizards naturally drop their tails to escape from predators. Studies have used lizard tail snips previously for host competency investigations and found *Borrelia burgdorferi sl* infections in lizards from Florida, Maryland, and South Carolina [22,23,24,42]. However, *Borrelia* bacteria can disseminate throughout the host body to multiple tissues, and the collection of tissue from additional internal organs could provide more insight on lizard infection status [23]. A laboratory study found that kidney and liver tissues of *P. inexpectatus* can be infected with *Borrelia burgdorferi ss* [23]. Our study used a combination of tail, liver, and skeletal muscle for our *Borrelia* assays so that we could detect Borrelia if it was disseminated in different parts of the lizard. In addition, our sample size was adequate; we sampled 228 lizards, yet a sample size of only 73 lizards was estimated as necessary to determine an apparent prevalence of 0.01 with a 95% CI. Because tick and lizard tissue samples tested negative, our results provide further evidence that lizards are unlikely to be important reservoir species for *Borrelia* in Florida. Future investigations should focus on other vertebrate hosts as potential competent hosts of *Borrelia*. 

Overall, we found that lizards are important hosts for *I. scapularis* but did not detect *Borrelia burgdorferi sl* in the tissues sampled. Tick–host association is an important factor in enzootic transmission of *Borrelia* bacteria. Previous investigations of Lyme disease transmission have shown that nymphal infection density (DIN) is a crucial indicator of disease risk [43,44]. Infected nymphs are important for infecting naive larvae and host species because *B. burgdorferi sl* species can be horizontally transmitted during cofeeding on a host [45]. In our study, we found both immature stages on lizards, but no *Borrelia burgdorferi sl* infections. Since both larvae and nymphs are feeding on potentially poor reservoir host (lizards), the DIN is low, which may reduce the risk of disease transmission [44]. These findings provide further evidence to support that the availability of competent reservoir hosts across the latitudinal gradient plays an important role in Lyme disease risk [6].

In addition to screening for *Borrelia burgdorferi sl,* we also surveyed our tick and lizard tissue samples for additional tick-borne bacterial pathogens. *Ixodes scapularis* is a known vector of other bacterial pathogens including *Anaplasma phagocytophilum* and *Ehrlichia muris euclairensis* [46]. From our tick samples, we only detected the presence of *Rickettsia* spp. which are likely endosymbionts based on GenBank matches to our sequences. Our lizard tissue samples all tested negative for bacterial pathogens, providing evidence that lizard species are unlikely reservoir species for other bacterial pathogens.

## 4. Materials and Methods

### 4.1. Museum Samples

Native lizard specimens from the Florida Museum of Natural History Herpetology Collection were examined for the presence and intensity of tick infestations. Eleven species of lizards were selected for examination based on previous host records and presence in Florida [47]. Lizard specimens were collected between 1902–2018 and collected from all 67 Florida counties. Lizards were examined under a dissecting scope to visualize any larval or nymphal ticks. Ticks were then carefully removed from reptiles using fine-tipped forceps and placed in vials containing 70% ethanol. Ticks were identified to species using taxonomic keys [48,49,50]. Based on the results of the museum survey, we targeted specific species of lizards in the field. 

### 4.2. Field Sampling of Lizard and Ticks

The presence of *Ixodes scapularis* has been reported in all Florida counties [26]. To incorporate the full distribution *of I. scapularis* we live sampled lizards from two field sites each in northern, central and southern Florida (N = 6) (Figure 1). Our 2 northern sites were near locations that previously detected *Borrelia burgdorferi sl* from lizards [22,24]. Based on the survey of museum specimens, we targeted *Plestiodon* skinks in our field study. Ticks were collected from lizards at the field sites from June to September 2020, which coincides with the *Plestiodon* spp. breeding season [25,51]. Each field site was sampled for a two-week period. Lizards at each site were captured using drift fences, cover boards, hand capture, or lizard fishing [52]. Collection methods were approved by University of Florida Institutional Animal Care and Use Committee (#201910934) and the Florida Fish and Wildlife Commission (FWC) (LSSC-20-00001). Captured lizards were visually examined for ticks using a magnifying glass. If ticks were present, they were removed with fine tip forceps and placed into vials with 100% molecular-grade ethanol. A 1 cm tail snip was taken from each lizard captured. Tails snips were placed in vials with 100% molecular grade ethanol. Lizards were subsequently released at the point of capture.

### 4.3. PCR Testing and Sequencing

To determine if lizards are potential reservoirs for bacterial pathogens, including *Borrelia* spp., we obtained 103 tissue samples from 11 native lizard species from the Genetic Repository of the Florida Museum of Natural History (Table 1). Tissue samples were collected between 2007–2018. Tissue sample types included liver (10), muscle (86), and tail tip (7). We also collected tail snips from 125 field-captured lizards. Ticks collected from captured lizards were pooled by individual lizard for DNA extraction. Tick specimens from the museum collection were not screened for pathogens since herpetological specimens are fixed with formalin which inhibits PCR.

All lizard tissue (museum and live capture) and field-collected tick samples were extracted for DNA using the Qiagen Gentra Purgene kit using the manufacture protocol (Qiagen, Valencia, CA, USA). Eluted DNA was stored at −20 ℃ until PCR was conducted. We ran PCR’s for 4 partial gene sequences of *Borrelia*: 16S, flab, IGS, and OspA [53,54,55]. We additionally tested for other tick-borne bacterial pathogens: *Anaplasma* spp., *Ehrlichia* spp. and *Rickettsia* spp. For *Anaplasma* and *Ehrlichia* we screened for the groESL gene [55]. For *Rickettsia* we screened with gltA which broadly screens for multiple species of *Rickettsia*. We also screened for Spotted Fever Group *Rickettsia* using the ompA gene [56]. The positive control for *Borrelia* used in our PCR was a construct of *B. burgdorferi ss* B-31 (ATCC® 35210™). For the *Anaplasma* positive control, a supernatant from dog cells infected with *A. phagocytophilum* was used. The *Rickettsia*-positive control was DNA from a *Rickettsia sp*. endosymbiont extracted from an *I. scapularis*. A negative control was run for each PCR assay with PCR-grade water. All PCR products were run on a 1.5% gel with RedView Stain (Genecopoeia, Rockville, MD, USA) and visualized on UVP gel documentation system (Analytik-Jena, Beverly, MA, USA). Positive samples were cleaned with SAP/Exonuclease and sent to Eurofins (Sanger Sequencing, Louisville, KY, USA) for sequencing to determine pathogen species. Sequences were aligned using Geneious (v 11.1) and compared to sequences in GenBank using NCBI BLAST.

## 5. Conclusions

In our investigation, we addressed two components of host competency: (1) Are *I. scapularis* ticks feeding on lizards in Florida? (2) Are lizards infected with *Borrelia burgdorferi sl*? Overall, we found that multiple lizard species in Florida are common hosts for *I. scapularis* immatures. Based on our study, *Plestiodon* skinks are common hosts of *I. scapularis* as evidenced by museum specimens and field-collected data from Florida. With multiple *Plestiodon* species distributed across the entire state, *I. scapularis* have hosts that can maintain immature life stages. We did not detect *Borrelia burgdorferi sl* or other tick-borne bacterial pathogens from ticks or lizard tissue samples. Our study provides further evidence that *Plestiodon* spp. are unlikely reservoir host species for tick-borne bacterial pathogens in Florida. 

## Figures and Tables

**Figure 1 pathogens-10-00999-f001:**
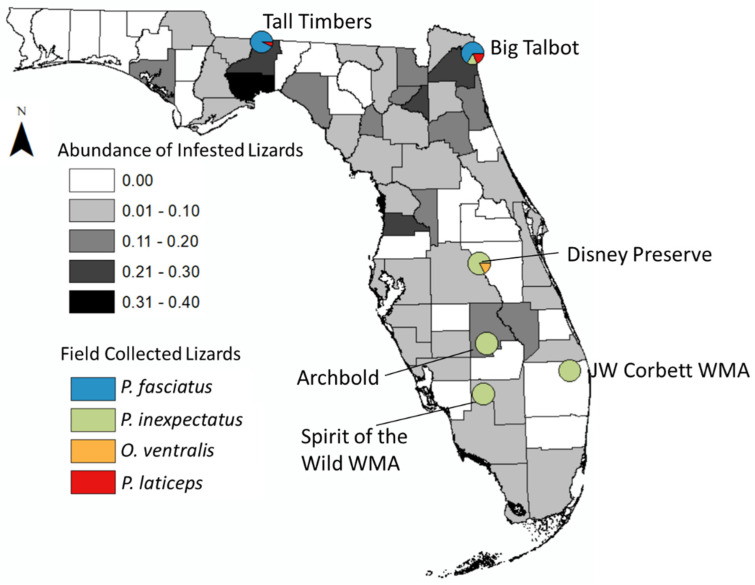
Map of field site locations and abundance of lizards infested with *I. scapularis*. Abundance estimates are based on combined museum and field sampling. Data were aggregated by county, and Florida counties are shaded from white to black based on the abundance of infested lizards (lizards with ticks/ total lizards sampled) from museum specimens and field collections examined from all 67 counties. Pie charts indicate the location of each field site and the proportion of each lizard species captured. WMA = Wildlife Management Area.

**Table 1 pathogens-10-00999-t001:** Lizard species surveyed for ticks and pathogens. The number in parentheses next to the species names represents the number of lizard tissue samples from the museum tissue repository tested for bacterial pathogens. Additionally, presented are the average number (±SE) of larvae and nymphs *I. scapularis* removed from individual herpetological specimens and the abundance (no. of lizards infested with ticks/no. of lizards sampled) and percentage of lizards infested with ticks.

Species	Avg. Larvae	Avg. Nymphs	Abundance of Infested Lizards (CI)
*Anolis carolinensis* (18)	0	0	0/1241 (0%) (±0.004)
*Aspidoscelis sexlineata* (2)	0	0	0/538 (0%) (±0.004)
*Plestiodon egregious* (10)	0	1	1/966 (0.01%) (±0.01)
*Plestiodon fasciatus* (4)	1.8 (±1.19)	2.1 (±1.62)	8/43 (18.6%) (±0.12)
*Plestiodon inexpectatus* (6)	4.7	1.2	125/753 (16.6%) (±0.03)
*Plestiodon laticeps* (6)	5.4 (±0.73)	2.2 (±0.33)	108/432 (25.0%) (±0.04)
*Ophisaurus attenatus* (7)	19.5	1.5	1/102 (0.98%) (±0.03)
*Ophisaurus compressus* (13)	0	0	0/89 (0%) (±0.04)
*Ophisaurus ventralis* (22)	7.1 (±1.82)	1.7 (±0.38)	24/381 (6.2%) (±0.03)
*Scincella lateralis* (10)	0.7 (±0.41)	0.4 (±0.25)	8/668 (1.2%) (±0.01)
*Sceloporus undulatus* (5)	2.5 (±0.79)	0.3 (±0.15)	11/615 (1.7%) (±0.01)

*Ophisaurus attenatus* and *Plestiodon egregius* only had 1 specimen infested with ticks so standard error could not be calculated. Wilson Score 95% confidence interval was calculated for abundance.

**Table 2 pathogens-10-00999-t002:** Abundance (no. of lizards infested with ticks/no. of lizards sampled) and percentage (in parentheses) of infested lizards at each field site. *Plestiodon fasciatus* and *P. laticeps* were not present in central and southern Florida and were left blank (-). WMA = Wildlife Management Area.

Location	*P. inexpectatus*	*P. fasciatus*	*P. laticeps*	# Ticks Collected
North				
Big Talbot	3/7 (42.9%)	24/32 (75.0%)	9/9 (100%)	137
Tall Timbers	0	24/40 (60.0%)	3/3 (100%)	89

Central				
Disney Preserve	1/10 (10.0%)	-	-	1
Archbold	4/16 (25.0%)	-	-	5

South				
JW Corbett WMA	0/1 (0%)	-	-	0
Spirit of the Wild WMA	1/4 (25.0%)	-	-	1
Total	9/38 (23.7%)	48/72 (66.7%)	12/12 (100%)	233

**Table 3 pathogens-10-00999-t003:** Average number of ticks removed from field-captured lizards (±SE) and abundance of tick-infested lizards. Abundance (no. of lizards infested with ticks/no. of lizards sampled).

Species	Sex	Avg. Larvae	Avg. Nymphs	Avg. All Ticks	Total Ticks	Abundance of Infested Lizards
*P. fasciatus*	Male	0.7 (±0.56)	3.4 (±0.56)	4.1 (±0.68)	74	18/25 (72.0%) (±0.17)
	Female	0.3 (±0.11)	3.1 (±0.58)	3.4 (±0.57)	78	23/31 (74.2%) (±0.15)
	Juvenile	0.1 (±0.14)	0.7 (±0.14)	1.0 (±0.0)	7	7/16 (43.8%) (±0.22)

*P. inexpectatus*	Male	0	3.5 (±0.75)	3.5 (±0.75)	7	2/17 (11.8%) (±0.15)
	Female	0.3 (±0.33)	1.3 (±0.33)	1.7 (±0.33)	5	3/10 (30.0%) (±0.16)
	Juvenile	0	0.8 (±0.50)	0.8 (±0.50)	3	4/12 (33.3%) (±0.24)

*P. laticeps*	Male	0	2.5 (±1.11)	2.5 (±1.11)	15	5/5 (100%) (±0.22)
	Female	1.1 (±0.59)	5.3 (±1.22)	6.4 (±1.55)	45	7/7 (100%) (±0.18)
	Juvenile	NA	NA	NA	NA	NA

All *P.* spp.		0.4 (±0.16)	3.0 (±0.30)	3.4 (±0.38)	233	69/123 (56.1%) (±0.09)

NA = No juveniles. Wilson Score 95% confidence interval was calculated for abundance.

## Data Availability

Data are available upon request to the corresponding author.

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
