# Peer review of "Reptile Host Associations of Ixodes scapularis in Florida and Implications for Borrelia spp. Ecology"

_pathogens, 2021, doi:10.3390/pathogens10080999_

Round 1

Reviewer 1 Report

Line 3: Borrelia spp.

Line 15: ... six ....

Line 18 and 19: Borrelia spp.

Line 127: Two tick species .....

Table 1 and Table 3: Too long description of the table contents ! Explanations to the table must be given below the table as "comments" or "explanations"

Line 173: .... the abudance (total of tick .....

Line 185: Ehrlichia spp., Anaplasma spp., ..........

Line 187, 188 and 189: spp.

Line 268: Xenodiagnostics ....

Line 284: .... infecting native larvae .....

Line 325: Borrelia spp.

Line 337, 338, 340 and 341: spp.

References: Standardize the notation of publication titles ! For example: not ... Biology and life History ..., but: ... biology and the history .....

Author Response

Reviewer 1 Comments and Suggestions:

Line 3: Borrelia spp. –

Corrected

Line 15: ... six ....

Corrected

Line 18 and 19: Borrelia spp.

Corrected

Line 127: Two tick species .....

Corrected

Table 1 and Table 3: Too long description of the table contents ! Explanations to the table must be given below the table as "comments" or "explanations"

Table descriptions were shortened.

Table 1: Moved text about specific calculations or missing data to below table.

Table 3: Information about no juveniles, standard error and 95% CI moved below table.

Line 173: .... the abundance (total of tick .....

Corrected

Line 185: Ehrlichia spp., Anaplasma spp., ..........

Corrected

Line 187, 188 and 189: spp.

Corrected

Line 268: Xenodiagnostics ....

Corrected

Line 284: .... infecting native larvae .....

Corrected

Line 325: Borrelia spp.

Corrected

Line 337, 338, 340 and 341: spp.

Corrected

References: Standardize the notation of publication titles ! For example: not ... Biology and life History ..., but: ... biology and the history .....

Citations have now been all standardized based on recommendations of MDPI reference guide

Author Response

General Comments: The manuscript describes research that helps clarify the lizard species that are important hosts for immature stages of Ixodes scapularis ticks in Florida. The information provided helps to build upon a growing body of evidence that lizards are retractile to infection with Borrelia burgdorferi the pathogen most likely to cause Lyme disease in North America. The authors describe an efficient way of focusing their field sampling design by using information obtained from preserved museum specimens. This reviewer has included specific information for the authors to consider in the

“Specific Comments” section below.

Specific Comments:                                                                                             

Page 1, Line 31: Suggest replacing “species in the complex” with “species in this complex”

Agreed. We have switched “ in the” to “ in this”.

Page 1, Line 37: The name “Borrelia” should be spelled out at the beginning of a sentence.

The sentence that starts on line 37 and goes to 38 starts with “ In California” and then goes to B. burgdorferi ss which is within the sentence.

We have ensured that when Borrelia is the first word in a sentence, it is spelled out completely. If B. burgdorferi is in the middle of a sentence, we have left it abbreviated. We will work with the copy editor to ensure that we are in compliance with the journal’s style for scientific name abbreviations.

Page 1, Lines 38-39: The sentence beginning on line 38 describing the prevalence of Lyme cases caused by B. burgdorferi 2 in the northeastern US vs the Midwest is incorrect. Borrelia burgdorferi is the most prevalent species to cause Lyme borreliosis in both the Midwest as well as the Northeast. In the Midwest, LB can also be caused by B. mayonii, in addition to B. burgdorferi. Please correct this sentence or provide references from the scientific literature that supports the claim as currently written.

The sentence has been corrected to reflect that B. burgdorferi is present in the Midwest and northeastern US while B. mayonii has only been detected in the Midwest. We removed the portion of the original statement about prevalence. The goal is to reflect that multiple B burgdorferi sl species are present depending on locality in the United States. The references cited accurately reflect the B burgdorferi sl species distributions.

Page 2, Line 64: Replace “Which” with “The” in the sentence beginning on line 64.

We removed “which” and replaced with “them”.

Page 2, Lines 76-77: The sentence beginning on line 76 “In California the western fence lizard plays a crucial role in the transmission of B. burgdorferi” is incorrect. Lizards do not transmit the pathogen that causes LB to people, Ixodes spp. ticks do. Lizards play a role in the ecology or maintenance of the bacteria in nature but not in its transmission. Please correct this sentence.

Agreed. We switched the word “transmission” to “ecology” since lizards in California do not transmit Borrelia.

Page 4, Lines138-139: The last sentence in this paragraph is incomplete, please edit.

This sentence has been edited to read “Ixodes scapularis were found on lizards which were collected as early as 1927 on P. lat-iceps specimens from Alachua County”.

Page 4, Figure 1: It is unclear from the Methods section (4.1) describing Museum samples if the 11 lizard species the authors focused on for their research were collected from all 67 counties in 3 Florida. If so please clarify in methods section, page 8, lines 298 to 303. If the 11 species the authors focused on for this research were not collected in all 67 FL counties then Figure 1 needs to be changed to indicate counties in which no lizard collections were performed in order to differentiate those counties from counties in which lizards were collected but no ticks were found on them, currently indicated as 0.0 in the Figure.

We agree and have edited this section for clarity. In addition we have edited the results section and the Fig 1 legend to maintain that clarity.

Also Figure 1 should be made larger in size, it is difficult to read the figure legends at its current size.

Figure 1 has been reworked for larger figure legends and increased overall size within the text.

Page 6, Line 195: Delete the second “the” in the first sentence of this paragraph.

Agreed. The second “the has been removed.

Page 6, Discussion section, Lines 197 to 296: The authors jump between the use of the terms “prevalence” and “abundance” when describing tick infestations on lizards, even though they defined abundance in Table 2 as the “# of lizards infested with ticks/# of lizards sampled”. The authors should stick with the term abundance and replace “prevalence” when used in its place in the Discussion section (and throughout the manuscript).

We have worked through the article to replace “prevalence” to abundance in reference to lizards infested with ticks.  We restrict the use of prevalence only to pathogen detection.

Page 6, Line 221: Replace “with” for “in” after the word “included”.

Agreed. We have replaced “in” and have changed it to “with”.

Page 6, Line 230: the word “tick” should be “ticks” in the sentence starting on line 230.

Agreed. We have added the plural “ticks”. Word is highlighted in text.

Page 7, Lines 283-284: It is unclear to this reviewer how infected nymphal ticks play a role in infecting naïve larval ticks? Please clarify this statement and include references from the scientific literature that indicates this phenomenon occurs with the bacteria that cause LB.

Line 300: We added a statement about the role of co-feeding on B. burgdorferi sl transmission between larvae and nymphs.

Page 7, Line 294: replace “of” with “on” prior to the word GenBank on line 294.

Line 308: We have switched “of” to “on”.

Reviewer 3 Report

Line 27: the estimate of annual cases is very out-dated.  Recent estimates place cases at 300,000 to as much as 476,000 cases of Lyme disease/annum.  Relevant recent citations for this ought to be referenced.

Line 33: suggest phrasing change from "...all not..." to "...not all pathogenic..."  (a minor stylistic point)

Line 39: is B. mayonii really the PREVALENT species in the mid-West??? Is there good documentation of this?  More common than BB ss??

Line 46: perhaps some mention of Lyme disease in California/Pacific Northwest & British Columbia is warranted?  There are recent references about this matter, including diversity of strains/

Line 138:  "Nonetheless....glass lizards." is not a sentence.

Line 230: "Studying wildlife infested with tick requires...." Insert "s" into word "ticks"

Line 280-281: suggest inserting the following caveat - "..but did not detect Borrelia burgdorferi sl in the tissues sampled."

Line 294-295: certain pathogens were not tested for, in particular, babesia or theileri piroplasms, also bartonella species. 

Line 321: so field tested lizard tissues were restricted to tail-snip only.  Perhaps state regulators prohibited sacrificing these animals for more in depth testing of multiple tissues?

Line 326:  What was the date/era when specimens from the Genetic Repository of the Florida Museum of Natural History were collected?  Might be worth mentioning.

Lines 326-329: Were the tissues from field-collected lizards restricted to 'tail-snip'?  Or other tissues as well. It ought to be considered that the tissues sampled were limited and did not include CNS tissues, cardiac tissues, kidney tissues, skin, or bladder.  These might (or might not) have been more revealing of presence of borreliae than muscle and liver only.  There might be mention of this; e.g. there is room for more in depth studies of lizard tissues taking a more thorough and systematic approach to sampling from more organ systems than in the current study.

Over all, a very nice study, reflecting a great deal of work that is original observations and help to 'flesh out' the ecologic situation in Florida.  Criticisms are minor and aimed at improving mss. and/or suggesting future work to be done.

Might be some discussion or speculation about whether or not Florida lizards may have a 'sterilizing' effect vs. borreliae and if so what mechanism(s) might be operative?  Or mention specifically of the work of Lane on Western fence lizards and whether similar effects are observed in Florida lizards and if so, what the mechanisms might be.  Again, 'grist' for the 'mill' for future research.

Author Response

Line 27: the estimate of annual cases is very out-dated.  Recent estimates place cases at 300,000 to as much as 476,000 cases of Lyme disease/annum.  Relevant recent citations for this ought to be referenced.

Agreed. We have replaced the older estimate with an updated reference which was most recent Lyme disease case estimates for the US at 476,000 (Kugeler et al., 2021)

Line 33: suggest phrasing change from "...all not..." to "...not all pathogenic..."  (a minor stylistic point)

Corrected

Line 39: is B. mayonii really the PREVALENT species in the mid-West??? Is there good documentation of this?  More common than BB ss??

Lines 37-39: We have corrected the text to reflect that both Bbss and B. mayonii are present in the Midwest. Based on a recent study, B. mayonii has a lower prevalence than Bbss in the Midwest (Lehane, 2021). A citation for this information has also been added.

Line 46: perhaps some mention of Lyme disease in California/Pacific Northwest & British Columbia is warranted?  There are recent references about this matter, including diversity of strains/

Line 37-39: Agreed. A statement about California B. burgdorferi sl distribution has been added.

Line 138:  "Nonetheless....glass lizards." is not a sentence.

Removed

Line 230: "Studying wildlife infested with tick requires...." Insert "s" into word "ticks"

Corrected

Line 280-281: suggest inserting the following caveat - "..but did not detect Borrelia burgdorferi sl in the tissues sampled."

Added

Line 294-295: certain pathogens were not tested for, in particular, babesia or theileri piroplasms, also bartonella species.

These pathogens were not tested for since there is currently no evidence to support that reptiles play a role in their transmissions cycles. We do specify we only tested for bacterial pathogens within the text.

Line 321: so field tested lizard tissues were restricted to tail-snip only.  Perhaps state regulators prohibited sacrificing these animals for more in depth testing of multiple tissues?

No, we did  samples 96 liver or skeletal muscle tissue types from the Florida Museum Genetic Repository and did not find any positive samples.  Our animal protocol was for live-sampling only and thus tail snips were the only tissue type for field collected specimens. We have made additional reference to the muscle and liver sampling in the discussion, beginning on line 288.

Lines 326-329: Were the tissues from field-collected lizards restricted to 'tail-snip'?  Or other tissues as well. It ought to be considered that the tissues sampled were limited and did not include CNS tissues, cardiac tissues, kidney tissues, skin, or bladder.  These might (or might not) have been more revealing of presence of borreliae than muscle and liver only.  There might be mention of this; e.g. there is room for more in depth studies of lizard tissues taking a more thorough and systematic approach to sampling from more organ systems than in the current study.

We agree that a more thorough study of organ systems is needed and have made mention of this in lines 285-288.

Line 326:  What was the date/era when specimens from the Genetic Repository of the Florida Museum of Natural History were collected?  Might be worth mentioning.

Lines 317: We have added collection date range for genetic repository samples

Might be some discussion or speculation about whether or not Florida lizards may have a 'sterilizing' effect vs. borreliae and if so what mechanism(s) might be operative?  Or mention specifically of the work of Lane on Western fence lizards and whether similar effects are observed in Florida lizards and if so, what the mechanisms might be.  Again, 'grist' for the 'mill' for future research.

Lines 270-280:  We agree and have added several sentences that describe Borreliacidals in lizards of the southeast and mention that this aspect requires further investigation.

Reviewer 4 Report

Manuscript untitled „Reptile host-association of Ixodes scapularis in Florida and implications for Borrelia ecology” is a result of novel research research on endemic areas for ticks and tick-borne diseases in United States of America. This kind of research has epidemiological as well as in veterinary and human field. Lizards are common in environment and presented studies meaning of ticks and tick-borne disease in wild environment of USA.

Article is divided into clear parts and very organized in logic way. In introduction we have a nice presented background about lizards and they connections with ticks epecially with Ixodes scapularis. In material and methods sector all procedures, areas of collecting samples and research techniques are described in details. Results are very well described but graphical presentation might be better performed. More tables ang graphs will makes this paper proofed.

Discussion is full with information about already discovered knowledge and in nice way compare with obtained results. As a conclusion authors emphasized that they found a diversity of lizards which are hosts for Ixodes scapularis, but simultaneously not only Borrelia burgdorferii sl was not found in ticks from lizards as well as in lizards samples but also other common tick-borne pathogens such as Ehrlichia, Anaplasma and Rickettsia were not found by ising molecular biology technique.

To sum up, I give a positive opinion about manuscript untitled Reptile host-association of Ixodes scapularis in Florida and implications for Borrelia ecology”.

Author Response

Manuscript untitled „Reptile host-association of Ixodes scapularis in Florida and implications for Borrelia ecology” is a result of novel research research on endemic areas for ticks and tick-borne diseases in United States of America. This kind of research has epidemiological as well as in veterinary and human field. Lizards are common in environment and presented studies meaning of ticks and tick-borne disease in wild environment of USA.

Article is divided into clear parts and very organized in logic way. In introduction we have a nice presented background about lizards and they connections with ticks epecially with Ixodes scapularis. In material and methods sector all procedures, areas of collecting samples and research techniques are described in details. Results are very well described but graphical presentation might be better performed. More tables ang graphs will makes this paper proofed.

Discussion is full with information about already discovered knowledge and in nice way compare with obtained results. As a conclusion authors emphasized that they found a diversity of lizards which are hosts for Ixodes scapularis, but simultaneously not only Borrelia burgdorferii sl was not found in ticks from lizards as well as in lizards samples but also other common tick-borne pathogens such as Ehrlichia, Anaplasma and Rickettsia were not found by ising molecular biology technique.

To sum up, I give a positive opinion about manuscript untitled „Reptile host-association of Ixodes scapularis in Florida and implications for Borrelia ecology”.

Response: Thank you for the positive opinion. We did not add any additional tables given the number that were already in the paper.